# HART: Efficient Adaptation via Regularized Autoregressive Parameter Generation

## Abstract

Fine-tuning is an effective approach for adapting a pre-trained language model to downstream tasks, but it incurs a high computational cost. To achieve an extremely efficient task adaptation, Phang et al. (2022) have proposed to use an auxiliary hypernetwork to generate task-specific weights without any backpropagation. A hypernetwork can generate weights for parameter-efficient fine-tuning (PEFT) modules, such as prefixes (Li & Liang, 2021) and LoRAs (Hu et al., 2021), for any unseen task based on a few task-specific demonstration examples, at the cost of a single forward pass. However, hypernetwork training is challenging. Firstly, it is sample inefficient due to the under-exploitation of the dependencies between PEFT weights across layers. Secondly, it exhibits training instability due to the high diversity of few-shot demonstration inputs. To address these limitations, we propose a novel hypernetwork training approach, named HART. It exploits layerwise dependencies by autoregressively generating weights for individual layers, and stabilizes the training by regularizing the consistency between weights generated based on different demonstrations. We train the hypernetwork on a diverse collection of tasks (Wang et al., 2022b; Sanh et al., 2021) and evaluate its performance on unseen tasks. HART notably outperforms Phang et al. (2022) on both T5-Large and T5-XL models.

## 1 Introduction

Pre-trained large language models (LLMs) have demonstrated remarkable capabilities on various tasks (Zhang et al., 2022a; Ouyang et al., 2022; Bubeck et al., 2023; Touvron et al., 2023; Wei et al., 2022a). While fine-tuning is an effective approach for adapting pre-trained models to specific tasks, it incurs significant computational costs, which further escalates with the increasing model size. In contrast, in-context learning (ICL) can quickly generalize a pre-trained model to unseen tasks by conditioning the model's inference on a few demonstration examples (Wang et al., 2022a; Wei et al., 2022b; Zhou et al., 2022; Xie et al., 2021; Min et al., 2022). However, without fine-tuning, the model weights lack the adaptability to each task.

To achieve an extremely fast adaptation to unseen tasks, Phang et al. (2022) have proposed a hypernetwork approach. A hypernetwork is a text-to-weight generator, which learns a universal mapping across a large collection of tasks from few-shot demonstration examples to parameter-efficient fine-tuning (PEFT) module weights of a pre-trained model, such as the weights of prefixes (Li & Liang, 2021), LoRAs (Hu et al., 2021) and adaptors (Houlsby et al., 2019). Consequently, when provided with several few-shot demonstration examples from an unseen task, the hypernetwork can generate the PEFT parameters for that task. Compared to iteratively fine-tuning task-specific parameters from scratch, generating these parameters requires only a single forward pass, thereby facilitating an extremely fast adaptation to various unseen tasks.

The hypernetwork adopts an encoder-decoder Transformer architecture, in which the decoder generates parameters conditioned on the encoded demonstration examples. To train such a hypernetwork, the decoder generates a hidden state at each training iteration, which is then projected, by different MLPs, into the weight spaces of different layers in the pre-trained model (referred to as "the main model"). Then the hypernetwork is optimized based on the main model's prediction loss on a training example.

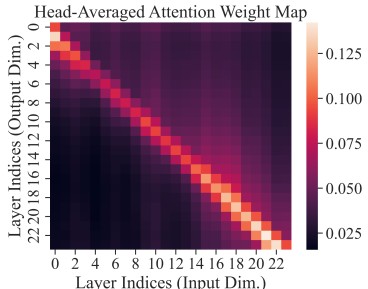

Figure 1: The bi-directional self-attention map averaged by attention heads in the last layer of the hypernetwork decoder. The input key/query is a sequence of hidden states, each responsible for learning PEFT parameters at a corresponding layer. Our experiment is conducted using T5-Large model (Raffel et al., 2020) on P3 (Sanh et al., 2021).

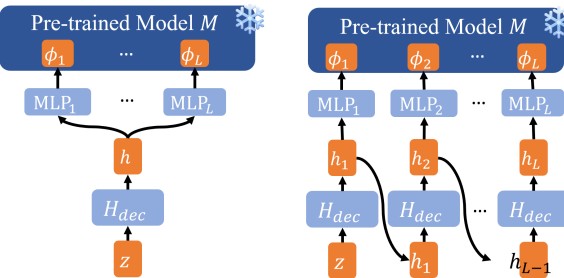

Figure 2: An illustrative comparison between non-autoregressive (left) and autoregressive (right) parameter generation schemes. $H_{dec}$: the hypernetwork decoder; $z$: a learnable input state; $h_l$: the hidden state at the $l$-th decoding step; $\phi_l$: the PEFT parameters for the $l$-th layer. $L$: the number of layer in the pre-trained model.

However, such a training scheme faces several limitations. One limitation is its low sample efficiency due to its heavy reliance on MLPs. Firstly, each MLP is assigned to learn parameters for a specific layer, based on a state that is shared among all layers. This implies that the entire responsibility of modeling the specificities of different layers fall upon the MLPs. However, the decoder, which has a greater capacity to capture layer-specific information in high-dimensional spaces, is not utilized to its fullest potential. Secondly, each MLP operates independently, rendering this design incapable of leveraging the potential structured dependencies between weights at different layers, which could serve as useful inductive biases during weight generation. In multi-layer networks, it is plausible to expect structured dependencies between weights across layers due to the structured dependencies between each layer's inputs: each layer only takes inputs from its preceding layer. To validate this hypothesis, we train the decoder to generate a sequence of hidden states using bi-directional attention, with each state responsible for learning the weights for an individual layer. Figure 1 presents the attention map across all states, revealing that each layer primarily attends to its two preceding layers, emphasizing its immediate predecessor. This observation confirms the existence of strong dependencies between weights at adjacent layers, an unexploited inductive bias for weight generation. Both the underutilization of the decoder's capabilities and underexploitation of the layerwise dependency impair the sample efficiency. As a result, the representation power of the generated parameters is compromised, leading to the underfitting of the main model.

To improve the sample efficiency, we propose a novel autoregressive parameter generation scheme as illustrated in Figure 2. Specifically, the decoder generates a sequence of hidden states autoregressively, with each state responsible for learning PEFT parameters of a corresponding layer. This allows each state to be learned through a layer-specific transformation modeled by the decoder. Furthermore, since each state is generated conditioned on its preceding state, the dependencies between adjacent layers are explicitly enforced, thereby introducing a proper inductive bias into the generation process. By capturing the layer-specificity and exploiting the layer-dependencies, the autoregressive scheme improves the expressiveness of the generated parameters.

Another challenge in hypernetwork training is its instability, which arises from the high diversity of inputs. As each input is a randomly selected sequence of demonstrations, considerable variation can occur across iterations, leading to significant variance in generated parameters. This variance destabilizes the main model's prediction loss and the hypernetwork training process (Figure 4). To resolve this issue, we propose a local consistency regularization method. This method discourages significant deviations in parameters generated between consecutive iterations, thereby stabilizing the gradient computation and the hypernetwork training process.

Finally, we propose HART, a novel **H**ypernetwork training scheme that incorporates both **A**uto**R**egressive parameter generation and local consistency regulariza**T**ion. We train the hypernetwork on diverse tasks and evaluate its generalizability on unseen tasks. Specifically, HART outperforms HyperTuning (Phang et al., 2022) by 1.6 points on the Super-NaturalInstructions task collection (S-NI, Wang et al. (2022b)) and by 3.6 points on the P3 task collection (Sanh et al., 2021) using the T5-XL model (Raffel et al., 2020). Additionally, thorough analysis confirms that HART generates well-fitted parameters and enhances the stability of hypernetwork training.

## 2 PRELIMINARIES

### 2.1 IN-CONTEXT LEARNING

**In-Context learning (ICL)** considers inferencing a pre-trained model on an unseen task based on a few demonstration examples (Liu et al., 2021; Wang et al., 2022a; Wei et al., 2022b; Zhou et al., 2022). Such an approach significantly outperforms zero-shot inference by leveraging the distribution of few-shot demonstrations as a task-specific prior for model prediction (Min et al., 2022; Xie et al., 2021). Specifically, the model takes a concatenated input of the few-shot demonstrations and a query to generate the response. Each input is written in a natural language template, such as a task-specific prompt or instruction, allowing the model to better interpret and follow the task intention.

**Multi-task In-Context Fine-tuning.** Due to the lack of weight adaptation, ICL significantly underperforms fine-tuning. To mitigate this gap, researchers have proposed to in-context fine-tune the pre-trained model on diverse training tasks before ICL (Min et al., 2021; Sanh et al., 2021; Wang et al., 2022b; Chung et al., 2022). Formally, we consider a pre-trained model denoted as $M(\cdot; \theta)$ parameterized by $\theta$, and a set of training tasks denoted as $\mathcal{T}$. At each training iteration, a task $\tau$ is sampled in proportion to the task size from $\mathcal{T}$. Then, a query denoted as $x$, its corresponding response denoted as $y$, and $K > 0$ demonstration input-output pairs denoted as $\{(x_k, y_k)\}_{k=1}^K$ are sampled from the task distribution $\mathcal{D}^\tau$. The model is optimized based on the following objective:

$$\min_\theta \mathbb{E}_{\tau \in \mathcal{T}, \{(x_k, y_k)\}_{k=1}^K, (x,y) \sim \mathcal{D}^\tau} \ell(M([d:x]; \theta), y), \tag{1}$$

where $\ell$ denotes the task loss and $d := [x_1 : y_1 : ... : x_K : y_K]$ denotes a concatenation of $K$-shot demonstrations. In prompted fine-tuning, $x, y$ and $d$ will be mapped to the prompted forms (Sanh et al., 2021; Min et al., 2021). In instruction fine-tuning, $d$ will be appended to a task definition (Wang et al., 2022b; Chung et al., 2022).

### 2.2 HYPERNETWORK

**Parameter Efficient Fine-tuning (PEFT)** introduces a minimal set of parameters, termed "PEFT parameters", to each layer of the pre-trained model and fine-tunes only these parameters while keeping the pre-trained model frozen (Li & Liang, 2021; Hu et al., 2021; Houlsby et al., 2019). Although PEFT attains adaptability comparable to standard fine-tuning (Ding et al., 2022), it incurs significant costs from full-model gradient backpropagation.

**Hypernetworks.** A hypernetwork is an auxiliary model trained to generate weights for a main model (Stanley et al., 2009; Ha et al., 2016). Phang et al. (2022) have proposed to use a hypernetwork to generate PEFT parameters for a pre-trained model. With a few demonstration examples from a new task, the hypernetwork can generate task-specific parameters, enabling the main model to conduct zero-shot inference on task-related queries. An advantage of this method is that, in contrast to PEFT, task adaptation requires only one forward pass, thereby cutting the backpropagation costs. When weighted against ICL, this method translates context into PEFT parameters, supporting zero-shot inference and improving task adaptability through task-specific weight adjustments.

The hypernetwork is trained in the multi-task in-context fine-tuning setting. Specifically, we denote a hypernetwork parameterized by $\xi$ as $H(\cdot; \xi)$. At each training iteration, the hypernetwork takes the $K$-shot demonstrations $d$ as input and generates the PEFT parameters. The main model takes in the query $x$ and generates the response $y$ using the PEFT parameters. The hypernetwork is then optimized based on the main model's prediction loss:

$$\min_\xi \mathbb{E}_{\tau \in \mathcal{T}, \{(x_k, y_k)\}_{k=1}^K, (x,y) \sim \mathcal{D}^\tau} \ell(M(x; \theta, H(d; \xi)), y). \tag{2}$$

**Parameter Generation in Hypernetworks.** The hypernetwork employs a Transformer encoder to encode the demonstrations and a Transformer decoder to conditionally generate parameters based on the encoded demonstrations. Specifically, we denote the decoder as $H_{\text{dec}}(\cdot; \xi_{\text{dec}})$. At the $t$-th training iteration, the decoder takes in a learnable token $z^{(t)} \in \mathbb{R}^{1 \times d_{\text{m}}}$, where $d_{\text{m}}$ is the hypernetwork's hidden dimension. It then generates a hidden state $h^{(t)} \in \mathbb{R}^{1 \times d_{\text{m}}}$ through one decoding step:

$$h^{(t)} = H_{\text{dec}}(z^{(t)}; \xi_{\text{dec}}^{(t)}, e^{(t)}), \tag{3}$$

where $e^{(t)}$ denotes the encoded demonstrations[1]. The generated hidden state is then projected into $L$ sets of PEFT parameters through $L$ learnable MLP layers:

$$\phi_l^{(t)} = \text{MLP}_l(h^{(t)}) \quad \forall l \in [L], \tag{4}$$

where $\phi_l^{(t)} \in \mathbb{R}^{d_\text{P}}$ denotes the PEFT parameters generated for the $l$-th layer of the main model, which consists $L$ layers.

## 3 METHOD

The parameter generation scheme in hypernetworks underexploits the layerwise dependencies and exhibits training instability. To address these issues, we propose HART, which exploits layerwise dependencies through autoregressive parameter generation and stabilizes training through local consistency regularization.

### 3.1 AUTOREGRESSIVE PARAMETER GENERATION

We propose to autoregressively generate the PEFT parameters for different layers in the main model. At the $t$-th training iteration, the hypernetwork decoder generates $L$ hidden states through $L$ decoding steps. Each state is generated conditioned on its preceding state, and is responsible for learning the weight space of the corresponding layer. At the first decoding step, the decoder takes in a learnable token, $z^{(t)}$, and generates the first hidden state, $h_1^{(t)}$, following Eq. 3:

$$h_1^{(t)} = H_\text{dec}(z^{(t)}; \xi_\text{dec}^{(t)}).$$

At the second decoding step, the decoder takes in $h_1^{(t)}$ and generates the second hidden state, $h_2^{(t)}$. This procedure is repeated for $L$ decoding steps:

$$h_l^{(t)} = H_\text{dec}(h_{l-1}^{(t)}; \xi_\text{dec}^{(t)}) \qquad \text{for } l = 2, ..., L.$$

Then we learn $L$ MLP layers to project these $L$ hidden states into $L$ sets of PEFT parameters, respectively, following Eq. 4:

$$\phi_l^{(t)} = \text{MLP}_l(h_l^{(t)}) \quad \forall l \in [L],$$

where $\phi_l^{(t)}$ denotes the set of PEFT parameters generated for the $l$-th layer of the main model. We can then compute the main model's prediction loss as:

$$\mathcal{L}_\text{pred}(\xi^{(t)}) = \ell(M(x^{(t)}; \theta, \{\phi_l^{(t)}\}_{l=1}^L), y^{(t)}). \tag{5}$$

where $\ell$ is defined in Eq 2.

The autoregressive generation scheme allows us to leverage a powerful decoder, instead of relying on MLPs, to model layer-specific transformations. Furthermore, this scheme exploits the strong dependencies between weights at adjacent layers as observed in Figure 1, thereby introducing a proper inductive bias into the generation process.

By exploiting the decoder's capabilities and the layerwise dependencies, the autoregressive generation scheme achieves a greater sample efficiency than the original scheme. As a result, the generated parameters are more expressive, leading to a better-fitted main model.

### 3.2 LOCAL CONSISTENCY REGULARIZATION

We further propose to encourage the PEFT parameters generated at consecutive training iterations to not deviate significantly from each other. At the $t$-th iteration, we compute the local consistency loss as:

$$\mathcal{L}_\text{cst}(\xi^{(t)}) = \text{MSE}([h_1^{(t)} : \ldots : h_L^{(t)}], [h_1^{(t-1)} : \ldots : h_L^{(t-1)}]),$$

where $\ell_\text{cst}(\xi^{(1)}) = 0$, $\text{MSE}(\cdot, \cdot)$ denotes the mean squared error, and $[h_1^{(\cdot)} : \ldots : h_L^{(\cdot)}] \in \mathbb{R}^{L \times d_m}$ is the concatenation of the sequence of generated hidden states. Finally, we optimize the hypernetwork based on the sum of the prediction loss and the consistency loss using an SGD-type algorithm:

$$\xi^{(t+1)} \leftarrow \xi^{(t)} - \nabla_{\xi^{(t)}}(\mathcal{L}_\text{pred}(\xi^{(t)}) + \mathcal{L}_\text{cst}(\xi^{(t)})),$$

---

[1]We omit $e^{(t)}$ for the rest of the paper to simplify the notations.

where $\mathcal{L}_{\text{pred}}(\xi^{(t)})$ is defined in Eq. 5.

By encouraging consistency between states generated at consecutive iterations, we mitigate drastic changes in the generated parameters arising from highly diverse inputs. This improves the smoothness of the main model, stabilizing both gradient computations and hypernetwork training.

## 4 EXPERIMENTS

We evaluate HART on several commonly used large-scale multi-task NLP benchmarks.

### 4.1 SUPER-NATURALINSTRUCTIONS (S-NI)

**Dataset.** Super-NaturalInstructions (S-NI, Wang et al. (2022b)) consists of 1616 tasks spanning 76 diverse categories, including translation, question answering and sentiment analysis, etc. Each task is associated with an expert-written task definition, a set of input-output pairs as positive and negative demonstrations, and a set of input-output pairs as training queries and responses. For each task, we construct the input to the hypernetwork as the concatenation of the task definition and two fixed positive demonstrations, denoted as "Def+2Pos". This input format has been observed to outperform other formats (Wang et al., 2022b). Following Phang et al. (2022), we select the English tasks for training and evaluation. We evaluate the generation performance of the main model on the held-out test set using the ROUGE-L metric.

**Model initialization.** All our experiments are conducted based on the LM-adapted T5 models (version 1.1, Lester et al. (2021)). T5 models are encoder-decoder Transformer-based models pre-trained using web-scale text-to-text corpus (Raffel et al., 2020)[2]. We consider two model scales: T5-Large (770M) and T5-XL (3B). For experiments on the T5-Large/XL, we initialize both the hypernetwork and the main model with the T5-Large/XL unless otherwise stated. We only keep the first 8 decoder layers (out of 24) in the hypernetwork for training efficiency. We randomly initialize the MLP layers.

**Training.** We freeze the main model and multi-task fine-tune the hypernetwork. The hypernetwork takes in input in the Def+2Pos format and generates the prefixes for the layerwise key and value representations (Li & Liang, 2021). We further adopt a **fusion-in-decoder** strategy, originally designed for question answering tasks (Izacard & Grave, 2020; Ye et al.). Ivison et al. (2022) has validated its effectiveness for hypernetworks, where the main model's decoder attends to the concatenated outputs from both the hypernetwork's encoder and the main model's encoder.

We fine-tune the hypernetwork for 10k steps in T5-Large experiments and 20k steps in T5-XL experiments. We use the Adam-8bit optimizer (Kingma & Ba, 2014; Dettmers et al., 2021) with a learning rate of $5 \times 10^{-5}$ and a batch size of 256. We select $\alpha \in \{1, 10, 20\}$. Further implementation details are deferred to Appendix 8.1.

**Inference.** For each task in the held-out test set, the hypernetwork takes in input in the Def+2Pos format and generates a set of prefixes. The main model then predicts all task queries using this single set of prefixes and the fused output from both encoders.

**Full Fine-tuning Baselines.** We list as references the baselines for fine-tuning the full model:
• **FT-Zero-Shot.** We multi-task fine-tune a model, which predicts the response to a query.
• **FT-Few-Shot.** We multi-task in-context fine-tune a model, which takes the concatenation of the task definition, two fixed positive demonstrations and a query as input, and predicts the response.
• **T$k$-Instruct** (Wang et al., 2022b) mainly differs from **FT-Few-Shot** in that each positive demonstration is further followed by an expert-written explanation.
• **HINT** (Ivison et al., 2022) is a hypernetwork approach where the hypernetwork and the main model share weights and are jointly fine-tuned. It consists of two stages: 1) The hypernetwork is pre-trained on the C4 corpus (Raffel et al., 2020). Each input string is split into three random-length chunks. The hypernetwork takes the first chunk as input to generate PEFT parameters. The main model takes the second chunk as input to predict the third chunk. 2) The hypernetwork and the main model share weights and are jointly in-context multi-task fine-tuned.

**Parameter Efficient Fine-tuning (PEFT) Baselines.** We compare with baselines for fine-tuning the PEFT parameters:

---

[2]LM-adapted T5 models further improve T5 models in activation function, dropout, parameter sharing and data filtration. We will omit "LM-adapted" when referring to the T5 models for the rest of the paper.

• **LoRA.** Hu et al. (2021) propose to add a pair of rank-decomposition weight matrices to each attention weight matrix and only fine-tune these matrices. We apply LoRA to **FT-Few-Shot**.

• **Prefix-Tuning.** Li & Liang (2021) propose to prepend a prefix to each key and value representation in attention modules and only train the prefixes. We apply Prefix-Tuning to **FT-Few-Shot**.

• **HyperTuning-PT** Phang et al. (2022) is a hypernetwork approach where the hypernetwork is fine-tuned while the main model is frozen. Similar to HINT, HyperTuning also consists of two stages: 1) The hypernetwork is pre-trained on the C4 corpus ("PT" stands for pre-training). Each input string is split into four chunks of predefined length. The hypernetwork takes the first and fourth chunk as input to generate PEFT parameters. The main model takes the second chunk as input to predict the third chunk. 2) The hypernetwork is in-context multi-task fine-tuned (Eq. 2).

• **HyperTuning** is our re-implementation of HyperTuning-PT where we remove the hypernetwork pre-training and adopt the fusion-in-decoder strategy.

**Comparison of the Hypernetwork Approaches.** Table 1 summarizes the differences between HART, HyperTuning-PT and HINT. Compared with HINT, HART unties the weights between the hypernetwork and the main model. While weight-sharing fine-tunes both models jointly and therefore achieves performance close to full fine-tuning, weight-untying can search for better PEFT parameters through leveraging a stronger hypernetwork (as we will see in Table 2). Furthermore, weight-freezing retains the benefits of conventional PEFT methods, e.g., it prevents catastrophic forgetting and saves the storage cost.

Compared with HINT and HyperTuning-PT, HART introduces autoregressive parameters generation and local consistency regularization. HART removes the hypernetwork pre-training to accommodate the computational budget, and adds the fusion-in-decoder approach to alleviate the resulting performance degradation (Table 6). We remark that the hypernetwork pre-training would be complimentary to HART.

Table 1: Comparison of HINT, HyperTuning-PT and HART.

| Method | Untie weights | Autoregressive generation | Pre-train hypernetwork | Adopt Fusion -in-decoder |
|---|---|---|---|---|
| HINT (Ivison et al., 2022) | No | No | Yes | Yes |
| HyperTuning-PT (Phang et al., 2022) | Yes | No | Yes | No |
| HART | Yes | Yes | No | Yes |

**Main Results.** Table 2 and Table 3 show the evaluation results of the T5-Large and T5-XL main models on the S-NI held-out test set, respectively. HART achieves an improvement of 1.6 points over HyperTuning in both the T5-Large and T5-XL experiments, demonstrating the effectiveness of autoregressive decoding and consistency regularization strategies. Compared with Prefix-Tuning and LoRA, HART achieves around 3 points of gain, suggesting that the PEFT parameters learned by HART are more generalizable than those learned by conventional PEFT methods.

We further use the Flan-T5-Large as the initialization to train the hypernetwork (the main model is still initialized as a T5-Large model). Flan-T5-Large is a T5-Large model instruction fine-tuned on the Flan collection (Chung et al., 2022), a more comprehensive and diverse task collection than SN-I. By leveraging a stronger hypernetwork, the performance of the main model increases by 2.8 points, outperforming the best full fine-tuning baseline by 1.6 points. This suggests that we can leverage the capability of a stronger auxiliary model to generate more expressive PEFT parameters than those could possibly be learned based on the main model.

## 4.2 PUBLIC POOL OF PROMPTS (P3)

**Dataset.** Public Pool of Prompts (P3, Sanh et al. (2021)) is a collection of prompted English datasets covering 62 tasks. Each task consists of a set of input-output pairs formatted in manually-written prompt templates. P3 was originally collected for the zero-shot setting, so it does not contain a demonstration set. Therefore, at each iteration, we sample 16 prompts from the training set and concatenate them to form the hypernetwork input, denoted as "16-Shots". We evaluate the main model on the held-out validation set using the multiple-choice scoring of accuracy. All model initialization, training and inference configurations follow Section 4.1.

**Full Fine-tuning Baselines: FT-Zero-Shot**, **FT-Few-Shot** and **HINT**. In **FT-Zero-Shot**, the input to the main model is a single prompted query. In **FT-Few-Shot**, the input is constructed as a concate-

Table 2: Evaluation results of the T5-Large main model on the S-NI held-out test set. For HART-FLAN-T5, we use the FLAN-T5-Large as the initial model to train the hypernetwork. Otherwise, we use the T5-Large as the initial model.

| Method | Avg. ROUGE-L |
|---|---|
| *Full Fine-tuning* | |
| FT-Zero-Shot | 40.6 |
| FT-Few-Shot | 47.6 |
| T*k*-Instruct (Wang et al., 2022b) | 48.0 |
| *Parameter-Efficient Fine-tuning (PEFT)* | |
| Prefix-Tuning (Li & Liang, 2021) | 42.6 |
| LoRA (Hu et al., 2021) | 42.9 |
| HyperTuning-PT (Phang et al., 2022) | 43.5 |
| HyperTuning | 45.2 |
| HART | 46.8 |
| HART-FLAN-T5-Large | 49.6 |

Table 3: Evaluation results of the T5-XL main model on the S-NI held-out test set. For all hypernetwork approaches, we use the T5-XL as the initial model to train the hypernetwork.

| Method | Avg. ROUGE-L |
|---|---|
| *Full Fine-tuning* | |
| FT-Zero-Shot | 46.6 |
| FT-Few-Shot | 54.0 |
| T*k*-Instruct (Wang et al., 2022b) | 54.3 |
| HINT (Ivison et al., 2022) | 53.2 |
| *Parameter-Efficient Fine-tuning (PEFT)* | |
| Prefix-Tuning(Li & Liang, 2021) | 47.1 |
| LoRA (Hu et al., 2021) | 47.7 |
| HyperTuning-PT (Phang et al., 2022) | 48.6 |
| HyperTuning | 48.8 |
| HART | 50.4 |

nation of 16 prompts and a prompted query. We also list the result of the **T0-3B** model as reported in Sanh et al. (2021), which was trained with **FT-Zero-Shot** on the complete P3 data collection.

**Parameter Efficient Fine-tuning (PEFT) Baselines: LoRA**, **Prefix-Tuning**, **HyperTuning-PT** and **HyperTuning**. All baselines adopt the 16-Shots input format.

**Main Results.** Table 4 and Table 5 show the evaluation results of T5-Large and T5-XL main models on the P3 held-out validation set, respectively. HART achieves 2.0 and 3.6 points of improvement over HyperTuning in the T5-Large and T5-XL experiments, respectively. However, as observed, HyperTuning underperforms HyperTuning-PT by 1.6 points in the T5-Large experiments and 5.6 points in the T5-XL experiments. We suspect that this is because the P3 collection is difficult to fit. In this case, the fusion-in-decoder approach becomes less effective as the encoder is too weak to extract representations that are meaningful, and hypernetwork pre-training becomes critical to facilitate the model convergence. Despite a lower baseline, HART still outperforms conventional PEFT methods and achieves performance comparable to the full fine-tuning baselines.

## 5 ANALYSIS

In this section, we justify the design choices for HART. All experiments herein utilize a T5-Large model for initializing both the hypernetwork and the main model.

### 5.1 ABLATION STUDY

Table 6 shows an ablation study of autoregressive parameter generation and local consistency regularization on the S-NI held out test set and the P3 held-out validation set. Starting with HyperTuning, we first remove the hypernetwork pre-training stage, then incorporate the fusion-in-decoder approach, and sequentially introduce the proposed strategies. We observe that autoregressive param-

Table 4: Evaluation results of T5-Large main model on the P3 held-out validation set. For all hypernetwork approaches, we use the T5-Large as the initial model to train the hypernetwork.

| Method | ANLI | HSwag | CB | COPA | RTE | WiC | WSC | WGD | Avg. |
|---|---|---|---|---|---|---|---|---|---|
| *Full Fine-tuning* | | | | | | | | | |
| FT-Zero-Shot | 33.4 | 28.0 | 63.0 | 77.9 | 71.1 | 50.8 | 61.0 | 53.4 | 54.8 |
| FT-Few-Shot | 35.3 | 27.5 | 68.6 | 70.5 | 75.2 | 51.7 | 62.1 | 52.2 | 55.4 |
| *Parameter-Efficient Fine-tuning (PEFT)* | | | | | | | | | |
| Prefix-Tuning (Li & Liang, 2021) | 33.1 | 26.1 | 53.9 | 67.8 | 60.5 | 49.8 | 54.7 | 51.4 | 49.7 |
| LoRA (Hu et al., 2021) | 31.8 | 26.3 | 48.6 | 61.4 | 71.3 | 51.5 | 63.0 | 51.1 | 50.6 |
| HyperTuning-PT(Phang et al., 2022) | 33.4 | 32.3 | 60.1 | 73.9 | 71.5 | 51.1 | 63.0 | 51.1 | 54.6 |
| HyperTuning | 33.4 | 28.5 | 59.4 | 68.6 | 67.9 | 50.6 | 62.8 | 52.9 | 53.0 |
| HART | 33.6 | 28.4 | 70.2 | 70.1 | 72.2 | 50.3 | 62.3 | 53.0 | 55.0 |

Table 5: Evaluation results of the T5-XL main model on the P3 held-out validation set. For all hypernetwork approaches, we use the T5-XL as the initial model to train the hypernetwork.

| Method | ANLI | HSwag | CB | COPA | RTE | WiC | WSC | WGD | Avg. |
|---|---|---|---|---|---|---|---|---|---|
| *Full Fine-tuning* | | | | | | | | | |
| T0-3B (Sanh et al., 2021) | 33.4 | 27.3 | 45.4 | 72.8 | 64.6 | 50.6 | 64.9 | 50.9 | 54.9 |
| FT-Zero-Shot | 39.9 | 29.4 | 64.5 | 88.0 | 80.8 | 51.7 | 60.7 | 57.9 | 59.1 |
| FT-Few-Shot | 37.9 | 30.9 | 67.6 | 90.5 | 76.6 | 51.2 | 63.3 | 61.1 | 59.9 |
| HINT (Ivison et al., 2022) | 41.6 | 30.3 | 76.0 | 88.8 | 84.2 | 51.4 | 59.5 | 60.1 | 65.4 |
| *Parameter-Efficient Fine-tuning (PEFT)* | | | | | | | | | |
| Prefix-Tuning (Li & Liang, 2021) | 38.3 | 31.2 | 61.4 | 82.4 | 78.6 | 52.6 | 57.0 | 54.3 | 57.0 |
| LoRA (Hu et al., 2021) | 37.3 | 27.7 | 54.2 | 77.5 | 74.6 | 53.9 | 58.3 | 51.6 | 54.4 |
| HyperTuning-PT (Phang et al., 2022) | 38.7 | 33.6 | 69.6 | 88.4 | 79.5 | 53.1 | 57.6 | 56.6 | 59.6 |
| HyperTuning | 36.8 | 26.6 | 54.6 | 79.4 | 76.8 | 52.3 | 54.8 | 50.8 | 54.0 |
| HART | 37.8 | 28.5 | 66.7 | 80.8 | 79.4 | 50.5 | 59.5 | 57.1 | 57.6 |

eter generation contributes over a one-point gain on both benchmarks, with local consistency adding an additional gain of approximately 0.5 points.

Table 6: Ablation study of the proposed components on the S-NI held out test set and the P3 held-out validation set.

| Method | S-NI Avg. ROUGE-L | P3 Avg. Score |
|---|---|---|
| HyperTuning-PT | 43.5 | 54.6 |
| - Continual Pre-training | 25.3 | 50.9 |
| + Fusion-in-Decoder (HyperTuning) | 45.2 | 53.0 |
| + Autoregressive Parameter Generation | 46.4 | 54.3 |
| + Local Consistency Regularization (HART) | 46.8 | 55.0 |

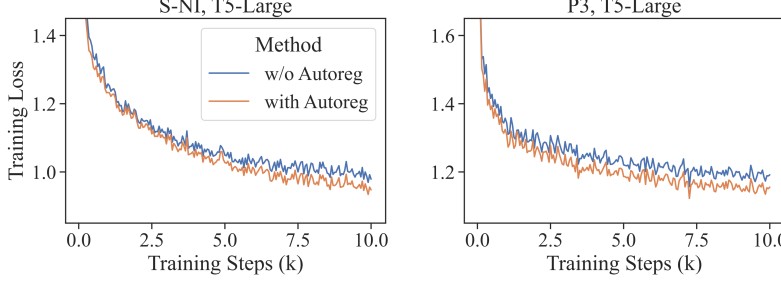

Figure 3: The main model prediction losses with and without using autoregressive parameter generation on the S-NI and the P3 training sets.

## 5.2 AUTOREGRESSIVELY GENERATED PARAMETERS FIT BETTER

Figure 3 shows the main model's prediction losses with and without autoregressive parameter generation on the S-NI and the P3 training sets. We present the loss curves corresponding to the experiments at the third and the fourth rows in Table 6, without regularizing the local consistency. By using autoregressively generated parameters, the training loss converges faster, suggesting that the parameters better fit the training data. Further analysis is deferred to Appendix 8.2.

## 5.3 LOCAL CONSISTENCY REGULARIZATION REDUCES LOSS VARIANCE

Figure 4 shows the main model's prediction losses with and without local consistency regularization the P3 training set. We present the loss curves corresponding to the experiments at the fourth and the last rows in Table 6. Local consistency regularization alleviates the loss spikes and reduces the loss variances, suggesting the training is more stable. Further analysis is deferred to Appendix 8.3.

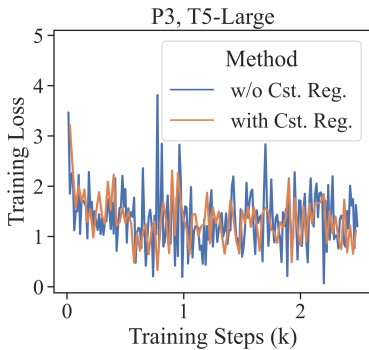

Figure 4: The main model prediction losses with and without using local consistency regularization on the P3 training set.

## 6 RELATED WORKS AND DISCUSSIONS

**Efficient Generalization to Unseen Tasks.** Recent research have explored efficient methods for adapting pre-trained LLMs to unseen tasks. One stream of research focuses on constructing inputs that can provide useful task-specific priors for model inference, including techniques such as prompt retrieval (Shi et al., 2023; Cheng et al., 2023), prompt optimization (Deng et al., 2022; Diao et al., 2022) and demonstration selection (Liu et al., 2021; Zhang et al., 2022b). While these methods bypass the costly weight updates, their performance is highly sensitive to the specific inputs used (Min et al., 2021). Another stream of research focuses on task-specific weight adaptation, exploring gradient-free methods as lightweight alternatives to gradient-based approaches like fine-tuning and PEFT. Chronopoulou et al. (2023); Huang et al. (2023) have proposed composing the weights of a new task by selecting relevant tasks and linearly combining their task-specific weights. While this approach is simple, it requires a pre-existing pool of task-specific weights, and the expressiveness of the composed weights may be constrained by the task pool. Phang et al. (2022); Ivison et al. (2022); Mu et al. (2023) have proposed using a meta model to generate task-specific weights or representations given related context. While the meta model may find more expressive weights, training such a meta model is challenging. Techniques like meta model pre-training (Phang et al., 2022; Ivison et al., 2022), and weight-sharing between the meta model and the pre-trained model (Ivison et al., 2022; Mu et al., 2023), are crucial for achieving training stability. Our work also aligns with this direction, offering new regularization scheme.

**Conditional Generation of DNN Weights.** Hypernetworks generally refer to the method of using one network to evolve another. The method draws inspiration from evolutionary computation, which suggests that massive brain connections exhibit structured patterns, representable by a composition of a small set of genotypes. Subsequently, existing works train hypernetworks as compositions of functions, each capturing specific weight structures, based on inputs containing information about weight structures (e.g., weight coordinates) (Stanley et al., 2009; Koutnik et al., 2010; Fernando et al., 2016; Ha et al., 2016). Generating high-dimensional weights in LLMs can be more challenging due to their complicated structures. In such cases, conditioning on information about weight structures can be more useful but is often overlooked in recent approaches. While our work exploits layerwise dependencies, there are more structures yet to be leveraged. For example, we may better initialize the mapping from the hypernetwork's latent space to the target weight space by pre-training it to predict the weight structures of existing LLMs.

## 7 CONCLUSION

To achieve an extremely efficient task adaptation, we propose a novel hypernetwork training approach, HART. HART incorporates an autoregressive decoding scheme to exploit layerwise dependencies and consistency regularization technique to improve training stability, allowing the hypernetwork to generate more expressive task-specific parameters for pre-trained models.

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

## 8 APPENDIX

### 8.1 IMPLEMENTATION DETAILS

#### 8.1.1 HYPERNETWORK ARCHITECTURE

The hypernetwork takes the encoder-decoder T5 Transformer architecture (Raffel et al., 2020). T5-Large and T5-XL models each consists of a 24-layer encoder and a 24-layer decoder. T5-Large has a hidden dimension of 1024 and T5-XL has a hidden dimension of 2048. For training efficiency, we adopt the first 8 layers of the decoder to initialize the hypernetwork's decoder.

#### 8.1.2 PARAMETER GENERATION IN DETAILS

In Sections 2 and 3, We introduce the parameter generation schemes for HyperTuning (Phang et al., 2022) and HART with some simplifications for presentation clarity. In this section, we provide the full details in their prefix generation schemes.

**Notations.** We denote the length of the prefix to be generated as $p$, which is set to be 32 in both methods. We denote the hidden dimension of both the hypernetwork and the main model as $d_\mathrm{m}$, which is 1024 in T5-Large and 2048 in T5-XL. We denote the number of layers in both the main model's encoder and decoder as $L$, which is 24 in both T5-Large and T5-XL.

**Parameter Generation in HyperTuning.** In HyperTuning, the input to the hypernetwork's decoder is a learnable embedding with $2p$ as the sequence length, denoted as $z \in \mathbb{R}^{2p \times d_\mathrm{m}}$. At each forward pass, the decoder takes in $z$ and generates a hidden state $h \in \mathbb{R}^{2p \times d_\mathrm{m}}$ using bi-directional self-attention following Eq. 3. The decoder conditions on the few-shot demonstration examples by crossly attending to the hypernetwork's encoder's output representation.

For each layer of the main model, two layer-specific MLPs would be learned to project the hidden state $h$ to the key and value prefixes, respectively. In other words, for a $2L$-layer main model, there are $4L$ MLPs. Each MLP consists of a layer normalization, 2 linear projections each with a dimension of $\mathbb{R}^{d_\mathrm{m} \times d_\mathrm{m}}$, and a tanh non-linear activation.

We denotes the MLP that learns the key/value prefix for the $l$-th layer of the encoder/decoder as $\mathrm{MLP}^l_\mathrm{enc,key}(\cdot)$, $\mathrm{MLP}^l_\mathrm{enc,value}(\cdot)$, $\mathrm{MLP}^l_\mathrm{dec,key}(\cdot)$, and $\mathrm{MLP}^l_\mathrm{dec,value}(\cdot)$, respectively. $\mathrm{MLP}^l_\mathrm{enc,key}(\cdot)$ and $\mathrm{MLP}^l_\mathrm{enc,value}(\cdot)$ would take $h[:p,:] \in \mathbb{R}^{p \times d_\mathrm{m}}$ as input and produce $\phi^l_\mathrm{enc,key}$ and $\phi^l_\mathrm{enc,value}$, both in $\mathbb{R}^{p \times d_\mathrm{m}}$, as the key and value prefixes for the $l$-th layer of the encoder. Similarly, $\mathrm{MLP}^l_\mathrm{dec,key}(\cdot)$ and $\mathrm{MLP}^l_\mathrm{dec,value}(\cdot)$ would take $h[-p:,:] \in \mathbb{R}^{p \times d_\mathrm{m}}$ as input and produce $\phi^l_\mathrm{dec,key}$ and $\phi^l_\mathrm{dec,value}$ as the key and value prefixes for the $l$-th layer of the decoder.

For training efficiency, $\{\mathrm{MLP}^l_\mathrm{enc,key}\}^L_{l=1}$ share the weights of their first linear projections. $\{\mathrm{MLP}^l_\mathrm{enc,value}\}^L_{l=1}$, $\{\mathrm{MLP}^l_\mathrm{dec,key}\}^L_{l=1}$ and $\{\mathrm{MLP}^l_\mathrm{dec,value}\}^L_{l=1}$ share their weights in a similar fashion.

**Parameter Generation in HART.** In HART, the input to the hypernetwork's decoder is a learnable embedding with 2 as the sequence length, denoted as $z \in \mathbb{R}^{2 \times d_\mathrm{m}}$. At each forward pass, the decoder takes in $z$ and autoregressively decodes a sequence of hidden states $h_1, ..., h_L$, each with dimension $\mathbb{R}^{2 \times d_\mathrm{m}}$. The decoder conditions on the few-shot task-specific demonstration examples by crossly attending to the hypernetwork's encoder's output representation.

For each layer of the main model, two layer-specific MLPs would be learned to project the hidden state $h$ to the key and value prefixes, respectively. Each MLP consists of a layer normalization, 2 linear projections and a tanh non-linear activation. The first linear projection is of dimension $\mathbb{R}^{d_\mathrm{m} \times p d_\mathrm{m}}$ and the second is of dimension $\mathbb{R}^{d_\mathrm{m} \times d_\mathrm{m}}$.

Following the same notations from HyperTuning, $\mathrm{MLP}^l_\mathrm{enc,key}(\cdot)$ and $\mathrm{MLP}^l_\mathrm{enc,value}(\cdot)$ would take $h[:1,:] \in \mathbb{R}^{d_\mathrm{m}}$ as input. After their first linear projections, the intermediate outputs are of dimension $\mathbb{R}^{p d_\mathrm{m}}$. We further reshape the intermediate outputs into the dimension $\mathbb{R}^{p \times d_\mathrm{m}}$, which is then projected by their second linear projections into $\phi^l_\mathrm{enc,key}$ and $\phi^l_\mathrm{enc,value}$, the key and value prefixes for the $l$-th layer of the encoder. Both prefixes are of dimension $\mathbb{R}^{p \times d_\mathrm{m}}$. Similarly, $\mathrm{MLP}^l_\mathrm{dec,key}(\cdot)$ and $\mathrm{MLP}^l_\mathrm{dec,value}(\cdot)$ would take $h[-1:,:] \in \mathbb{R}^{d_\mathrm{m}}$ as input and produce $\phi^l_\mathrm{dec,key}$ and $\phi^l_\mathrm{dec,value}$ as the key and value prefixes for the $l$-th layer of the decoder.

For training efficiency, $\{\mathtt{MLP}^l_{\mathrm{enc,key}}\}^L_{l=1}$ share the weights of their first linear projections. $\{\mathtt{MLP}^l_{\mathrm{enc,value}}\}^L_{l=1}$, $\{\mathtt{MLP}^l_{\mathrm{dec,key}}\}^L_{l=1}$ and $\{\mathtt{MLP}^l_{\mathrm{dec,value}}\}^L_{l=1}$ share their weights in a similar fashion.

**Remark regarding Weight Sharing and Input Sharing in MLPs.** We remark that in HyperTuning and HART, the MLPs are not completely independent across layers because they share the same input hidden state and the first linear projection layers. Such input and weight sharing indeed allow MLPs to learn the pattern of layerwise dependency through training, but they need to learn it from scratch. In contrast, autoregressive decoding allows the hypernetwork to directly exploit the layerwise pattern without learning. This pattern is an useful inductive bias that improves the sample efficiency during training.

### 8.1.3 TRAINING DETAILS

**Multi-task Training Data Sampling.** We follow the multi-task in-context fine-tuning setting from MetaICL (Min et al., 2021). At each training iteration, we first randomly sample a task from the training task pool, and then randomly sample $K$ shot demonstration examples and one training example from this task. During inference, for each task, we use a fixed set of demonstration examples for all test queries. For P3 training data, we exclude tasks with average sequence lengths longer than 320 tokens to fit more prompts into the input following Phang et al. (2022).

**Fusion-in-decoder.** We further adopt a fusion-in-decoder strategy, originally designed for question answering tasks (Izacard & Grave, 2020; Ye et al.). This strategy requires the decoder to attend to concatenated representations of multiple encoded input contexts. Ivison et al. (2022) has validated its effectiveness for hypernetworks. Specifically, at each forward pass, we prepend the hypernetwork's encoder output to the main model's encoder output, and require the main model's decoder to attend to such a fused representation in the cross attention module. This approach is adopted in both the training and inference stages.

**Hyperparameters.** We fine-tune the hypernetwork for 10k steps in T5-Large experiments and 20k steps in T5-XL experiments. For both model experiments, we use the Adam-8bit optimizer (Kingma & Ba, 2014; Dettmers et al., 2021) with a learning rate of $5 \times 10^{-5}$ and a batch size of 256. We adopt a linear decay learning rate schedule. We select $\alpha \in \{1, 10, 20\}$. We set the maximum input sequence length for the hypernetwork as $1024$ and the prefix length as $32$. For the main model, we set the maximum input and target sequence length as $384$ and $128$. We adopt the same input sequence length and target sequence length during inference.

We use deepspeed library for distributed training and inference. The T5-Large experiments are conducted on 8 Nvidia 32G V100 GPUs and T5-XL experiments are conducted on 8 Nvidia 80G A100 GPUs.

### 8.2 ADJACENT LAYERS EXHIBIT STRONGER DEPENDENCY

Table 7 shows the evaluation results under different decoding schemes: 1) generate a single layer-shared state (HyperTuning); 2) generate layer-specific states autoregressively for a randomly-chosen, fixed order of layers; 3) generate layer-specific states autoregressively from the top layer to the bottom layer; 4) generate layer-specific states autoregressively from the bottom layer to the top layer (HART). All experiments are conducted without regularizing the local consistency. We can observe that 2), 3) and 4) achieve noticeable improvements upon 1), demonstrating the benefit of utilizing the capabilities of the decoder. Furthermore, 5) outperforms 3), suggesting that exploiting the underlying problem structure improves the sample efficiency.

### 8.3 VISUALIZING CONSISTENCY REGULARIZED PARAMETERS

Figure 5 (Left) showcases the value of $\mathcal{L}_{\mathrm{cst}}$ with and without local consistency regularization on the S-NI training set. We can observe that the generated parameters no longer change drastically across iterations after applying local consistency regularization. Figure 5 (Right) shows the t-SNE plot of the hidden states generated for different layers with and without applying local consistency regularization on four P3 held-out test tasks. One observation is that, with or without regularization, the states generated for different tasks are clustered, while the states generated for different layers are scattered. This suggests that layer-specific weight structure maybe more distinct than task-specific weight structure. Another observation is that, after applying consistency regularization, the states generated for different layers become more diverse.

Table 7: Evaluation results of the T5-Large model on S-NI test set with PEFT parameters generated under different layerwise dependencies. *The result is obtained by averaging over three different randomly-chosen orders.

| Generate Multiple Layer-specific States? | Layerwise Dependency | S-NI Avg. ROUGE-L |
|---|---|---|
| No | N/A (HyperTuning) | 45.2 |
| Yes, autoregressively | Depend on one random, fixed layer* | 45.5 |
| Yes, autoregressively | Depend on the next layer | 45.8 |
| Yes, autoregressively | Depend on the previous layer (HART) | 46.4 |

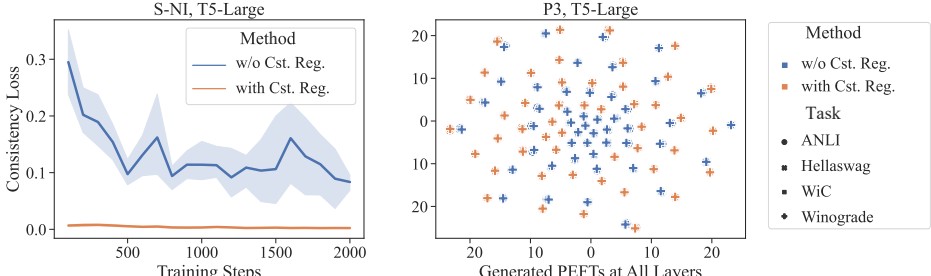

Figure 5: *Left*: The value of $\mathcal{L}_{cst}$ with and without local consistency regularization on the S-NI training set. *Right*: The t-SNE plot of the hidden states generated for different layers with and without applying local consistency regularization on four P3 held-out test tasks.

