# OpenReview forum: "HART: Efficient Adaptation via Regularized Autoregressive Parameter Generation"
_ICLR.cc/2024/Conference — Submitted to ICLR 2024_

### Official Review · Reviewer_Yxi9 · 2023-10-29

**Soundness:** 3 good
**Presentation:** 3 good
**Contribution:** 2 fair
**Rating:** 5
**Confidence:** 3

**Summary:**

The authors propose a novel hyper-network training scheme, which generates parameters for different layers using an autoregressive approach, thereby leveraging inter-layer dependencies and stabilizing the training process through local consistency regularization. They conduct experiments on multiple large-scale multi-task NLP benchmarks, demonstrating certain generalization abilities and efficiency.

**Strengths:**

- Using contextual conditions to generate parameters in hypernetworks is useful in practice, while the problem of low sample efficiency in parameter generation in hypernetworks is easily overlooked. The authors approach this issue from the perspective of parameter dependency between adjacent model layers. A simple yet effective parameter generation scheme is proposed, utilizing an autoregressive approach that enhances the expressiveness of the generated parameters.
- The paper conducts extensive experiments to demonstrate the effectiveness of the proposed method. The proposed method is compared with multiple strong baselines on the S-NI dataset consisting of 1616 tasks and the P3 dataset consisting of 62 tasks. The datasets with numerous tasks can better highlight how HART can generate task-specific parameters that are more expressive.
- The paper is well-organized and easy to follow.

**Weaknesses:**

- Due to the use of the T5 model's weights to initialize the hypernetwork, the structure of the hypernetwork is subject to certain constraints and cannot arbitrarily change the parameter sizes. At the same time, the initialization parameters of the hypernetwork (such as from T5-Large/XL) might significantly impact its performance. I am curious to see what the results would be if training a completely new hypernetwork from scratch.
- The large number of trainable parameters in the transformer-based hypernetwork contradicts the original intention of PEFT. Compared to the typical PEFT method, it may require more computational overhead.

**Questions:**

Please respond to the concerns listed in weaknesses.

---

> ### Author Response · Authors · 2023-11-19
>
> We appreciate the reviewer for providing the comprehensive and helpful reviews. We provide our responses below.
>
> **Due to the use of the T5 model's weights to initialize the hypernetwork, the structure of the hypernetwork is subject to certain constraints and cannot arbitrarily change the parameter sizes.**
>
> Using architectures and weights other than T5 to initialize the hypernetwork is flexible. We adopt T5 mainly because it is a widely used encoder-decoder model family for text-conditioned generation, offering multiple scale options.
>
> Furthermore, even sticking with a specific model, we can still change its parameter size. For example, in our experiments, we only use the first 8 layers (out of 24) in the T5 decoder for training efficiency.
>
> **The initialization parameters of the hypernetwork (such as from T5-Large/XL) might significantly impact its performance. I am curious to see what the results would be if training a completely new hypernetwork from scratch.**
>
> We agree with the reviewer that the hypernetwork initialization will impact its performance, but we emphasize that all methods in this paper are evaluated under the same initialization scheme, ensuring fair comparisons.
>
> Technically, we are able to train a hypernetwork from scratch but it will require much more resources than training from pre-trained weights. The pre-trained weights are valuable as they can provide the hypernetwork with the capability to understand the input demonstrations. So it makes sense to take advantage of pre-trained weights whenever they are available.
>
> **The large number of trainable parameters in the transformer-based hypernetwork contradicts the original intention of PEFT. Compared to the typical PEFT method, it may require more computational overhead.**
>
> We agree with the reviewer that training a hypernetwork is costly. However, for any new task, a trained hypernetwork can directly generate PEFT weights within one forward pass, which takes less than a second to finish. In contrast, PEFT requires iterations of forward and backward passes that take hours to finish. Note that the backward pass is not only slower than the forward pass but also requires more memory.
>
> Considering the case when the number of tasks scales up to $m$. The hypernetwork approach only requires $m$ forward passes, while PEFT requires $m \times n$ forward and backward passes, where $n$ is the number of training iterations for each task. As a result, the hypernetwork approach is more efficient than PEFT.
>
> ---
> We hope our responses address your concerns. We are happy to discuss your remaining concerns if any.

---

> > ### Comment · Reviewer_Yxi9 · 2023-11-22
> > **Response**
> >
> > Thank you for your response. I agree with the author's answer regarding pre-training weights. I will raise my score to 6. However, I still believe that during the fine-tuning phase, HART can directly generate PEFT weights in a single forward pass. However, before this, a large-scale hypernetwork still needs to be trained, which has certain limitations in terms of hardware cost. Additionally, in cases where pre-training weights are not available (such as in MLP-structured hypernetworks), the scalability of this method is not good.

---

### Official Review · Reviewer_cSGa · 2023-10-31

**Soundness:** 3 good
**Presentation:** 2 fair
**Contribution:** 2 fair
**Rating:** 6
**Confidence:** 3

**Summary:**

This work introduces a variation of the HyperNetwork-based weight generation. In contrast to previous approaches, the proposed method, known as HART, employs layer-wise sequential generation. To enhance the generation process, the authors have applied a local consistency regularization, which serves as a kind of smoothing term to ensure temporal consistency. Additionally, the proposed method incorporates a fusion-in-decoder strategy to operate effectively without the need for HyperPretraining. The authors have demonstrated the strong performance of this method even in scenarios where HyperPretraining is not utilized, specifically for T5-Large and T5-XL.

However, I have some concerns as follows:

1)	The level of novelty in this work appears limited. The main novelty relies on the adoption of layer-wise autoregressive generation. The local consistency regularizer can be considered a smoothing term or a momentum, and the use of a fusion-in-decoder strategy has been observed in various prior works. To underscore the novelty, it is crucial to conduct more in-depth analyses that concretely demonstrate the advantages of the layer-wise autoregressive model. Figure 3 only illustrates the training loss improvement achieved by the autoregressive model. It would be beneficial to compare the performance of each layer in autoregressive and non-autoregressive models.

2)	I think that the order of layers is another significant consideration for autoregressive method. While the authors may have utilized layer indices, these indices may not accurately represent the network architecture, particularly in cases involving skip connections or parallel layers. The layer index alone may not capture the sequential operations of the model.

3)	The analysis of local consistency could benefit from further exploration. Figures 4 (in the main manuscript) and Figure 5 (in the appendix) offer limited insight into the effects and importance of local consistency. Among them, the t-SNE plot appears to provide some insight, but there is a need for a more detailed explanation of the meaning behind distinct scattering patterns in weight space. A clarification of the physical significance of these patterns is recommended.

4)	In the equation for updating the hypernetwork, it is worth considering the inclusion of weighting factors between L_pred and Lcst. In cases where a model has numerous layers or a hypernetwork features high-dimensional hidden states, the update term may have varying impacts. Introducing normalization or weighting factors could help address this issue.

5)	While the proposed method exhibits significant performance improvements over previous works in the case of T5-Large, the gains are somewhat diminished in the case of T5-XL. It is important to note that this is not a critique of the performance itself, as it is understood that the proposed method performs better. However, it would be valuable to investigate the reasons behind this difference in performance. My guess is that T5-XL's increased number of layers may lead to challenges related to the layer-wise autoregressive model, such as forgetting or the layer index issue mentioned earlier. Addressing these potential weaknesses or limitations would be beneficial.

Overall, the proposed work demonstrates superior performance compared to previous works and holds practical utility. However, the main novelty lies in the layer-wise autoregressive model, and there is a need for more comprehensive analyses in this regard. I encourage the authors to provide additional insights during the rebuttal period.

**Strengths:**

See above

**Weaknesses:**

See above

**Questions:**

See above

---

> ### Author Response · Authors · 2023-11-19
>
> We appreciate the reviewer for reading through the paper carefully and providing the thoughtful reviews. We provide our responses below.
>
> **It is crucial to conduct more in-depth analyses that concretely demonstrate the advantages of the layer-wise autoregressive model... It would be beneficial to compare the performance of each layer in autoregressive and non-autoregressive models.**
>
> We are currently conducting an experiment that only generates a subset of layers autoregressively and checking if the performance would degrade. We hope to provide the results of this experiment before the rebuttal deadline.
>
> As all layers together deliver the final prediction performance, ***could the reviewer further clarify which specific "performance of each layer" the reviewer is interested in and what insights the reviewer is trying to obtain?*** We hope to provide useful analysis that addresses the reviewer's concerns.
>
> **While the authors may have utilized layer indices, these indices may not accurately represent the network architecture, particularly in cases involving skip connections or parallel layers. The layer index alone may not capture the sequential operations of the model.**
>
> We do not utilize layer indices. As shown in the second equation in Section 3.1, we directly take the hidden state generated for the $(l-1)$-th layer as the input to the decoder to generate the hidden state for the $l$-th layer. Therefore, the hidden state generated for the $l$-th layer depends on the hidden states generated for all layers before the $l$-th layer, with an emphasis on the $(l-1)$-th layer. Such dependencies are able to ***capture all skip connections*** in Transformer models, and therefore can accurately represent the network architecture.
>
> Note, ***there are no parallel layers*** in T5 models. Suppose there are $n$ parallel layers in some model: then we can always generate $n$ hidden states in parallel with a diagonal attention mask, rather than one hidden state, at every decoding step. Suppose there are parallel structures within the same layer: as elaborated in Appendix 8.1.2, these structures all stem from the same hidden state and are then projected by parallel MLPs.
>
> **Figures 4 (in the main manuscript) and Figure 5 (in the appendix) offer limited insight into the effects and importance of local consistency. Among them, the t-SNE plot appears to provide some insight, but there is a need for a more detailed explanation of the meaning behind distinct scattering patterns in weight space. A clarification of the physical significance of these patterns is recommended.**
>
> We suspect that the scattering pattern arises because consistency regularization encourages consistency across weights for ***different tasks***.
>
> Recall that, in multi-task learning, input at each training step is randomly sampled from all tasks, which means that $h_l^{(t)}$ and $h_l^{(t-1)}$ could be generated based on demonstrations from different tasks. Since $\mathcal{L}_{\rm cst}$ encourages the consistency between $h_l^{(t)}$ and $h_l^{(t-1)}$, it therefore encourages the consistency between weights across different tasks.
>
> In the absence of consistency regularization, the hypernetwork is trained to generate weights that ***overfit task-specific demonstrations***. The task-specific weights can be modeled in a specific weight space, so different layers generated for the same task are close.
>
> When the consistency regularization is applied, the hypernetwork is trained to generate weights that can ***generalize across tasks***. To search such weights, the hypernetwork needs to explore a larger weight space. As a result, different layers generated for the same task becomes more scattered. On the other hand, since the weights are task-shared, we can observe that the same layer generated for different tasks become closer.
>
> In summary, it is a "trade-off" between the cross-task and inner-task weight diversity. When the weights are more consistent across tasks, each task's weights tend to be more diverse across layers, and vice versa.
>
> By balancing such a trade-off, we are able to prevent the hypernetwork from generating weights that overfit to the demonstrations. This is important during inference: considering the cases where the few-shot demonstrations may misrepresent the actual data distribution of the unseen task (which could be common), we would want the hypernetwork to be robust to such noise.

---

> ### Author Response · Authors · 2023-11-19
>
> **In the equation for updating the hypernetwork, it is worth considering the inclusion of weighting factors between L_pred and Lcst.**
>
> The absence of the weighting factor in Section 3.2 is a typographical error. As listed in the "Hyperparameter" Section in Appendix 8.1.3, we consider a weighting factor $\alpha$ for $\mathcal{L}{\rm cst}$. We tune $\alpha$ in the range of {1, 5, 10, 20} in the S-NI experiments on T5-Large. We early stop the model at 50 percent of the total training steps and evaluate the Rouge-L score to be 44.1, 44.3, 44.3, and 44.4, respectively. We, therefore, set $\alpha=20$ across tasks and models without further tuning. At 10 percent, 50 percent, and 100 percent of the total training steps, the magnitude of $\alpha\mathcal{L}{\rm cst}$ is around 5 percent, 2 percent, and 1 percent of the total loss, respectively.
>
> **While the proposed method exhibits significant performance improvements over previous works in the case of T5-Large, the gains are somewhat diminished in the case of T5-XL... Addressing these potential weaknesses or limitations would be beneficial.**
>
> The gain of HART over HyperTuning is not diminished in the case of T5-XL: In Table 3 and 4, the gains are $1.6$ and $1.6$ for T5-Large and T5-XL, respectively. In Table 5 and 6, the gains are $2.0$ and $3.6$ for T5-Large and T5-XL, respectively.
>
> The reduced advantage of HART over HyperTuning-PT in the case of T5-XL can be attributed to the decreased advantage of HyperTuning over HyperTuning-PT. This occurs because Fusion-in-Decoder's effectiveness diminishes, and the importance of pre-training grows as the model size increases.
>
> ---
> We hope our responses address your concerns. We are happy to discuss your remaining concerns if any.

---

> ### Comment · Reviewer_cSGa · 2023-11-21
>
> Thank you for response.
>
> However, many concerns are not well addressed as follows:
>
> 1. Validation of 'Layer-wise Autoregressive Model"
>
>  What I'm specifically addressing is not the "performance of each layer," but rather the justification for using a layer-wise autoregressive model. The "performance of each layer" was just an example. I recognize that clarifying the layer-wise autoregressive model is challenging. Yet, as it represents the core innovation and contribution of the authors, its validation is essential.
>
> However, the paper lacks any substantive evidence supporting the "layer-wise" approach. While improvements in loss history and average ROUGE-L scores are presented, these alone do not suffice to substantiate the layer-wise autoregressive model's efficacy.
>
> As another reviewer noted, the proposed method seems to require more computational effort compared to previous models due to its layer-wise calculations. This raises a question: Are the observed improvements due to 1) the inherent advantages of layer-wise processing, or 2) simply a more complex model structure?
>
> In the absence of concrete proof, I am hesitant to rate this work highly. If I were in the authors' position, I would suggest analyzing various inputs during the decoding process (e.g., all hidden states from every layer, cumulative states from preceding layers, etc.) and visually demonstrating how these different inputs impact the model's output.
>
>
>
> 2. Sequential Processing for All Architectures
>
> I noted that the author incorporated the concept of a layer index, but this does not imply directly inputting the index into the hypernetwork. My point was that the proposed model employs the (l-1)-th hidden state as input for the decoder to generate the l-th hidden state. This approach, however, could pose challenges for parallel layer structures like skip connections.
>
> In the case where the (l+1)-th layer is a skip connection, the l-th and (l+1)-th layers run in parallel since their outputs are combined. According to the response, the (l-1)-th hidden states are inputs for the l-th layer. Thus, the l-th hidden state should be the input for decoding the (l+1)-th layer. However, here lies a discrepancy: the l-th layer isn't positioned before the (l+1)-th skip-connection layer, yet it's used in the decoding process.
>
> This indicates that the l-th layer isn't consistently preceding the (l+1)-th layer in the architecture, contradicting the response which states that the l-th layer relies on hidden states generated by all preceding layers. In my example, the (l-1)-th hidden state should be the input for both the l-th and (l+1)-th layers. Additionally, the authors claim that T5 models don't have parallel layers, yet T5 incorporates multi-head attention, which includes parallel fully-connected layers. The proposed method seems to use the hidden states of the "Value" as input for the "Key," and then those of the "Key" for the "Queue." Since T5 also includes skip connections, it appears the proposed method treats any model as sequential, ignoring specific architectural features. Moreover, Appendix 8.1.2, which describes parallel processing within the proposed method, doesn't address this concern.
>
>
>
> 3. Consistency Regularization addressing Task-specific Overfitting
>
> If the authors assert that "In the absence of consistency regularization, the hypernetwork is trained to generate weights that overfit task-specific demonstrations," they need to present clear evidence of such overfitting for substantiation. The provided materials, such as the scattering figure and marginally smooth loss depicted in Figure 4, are insufficient to validate their concerns about task-specific overfitting. More comprehensive data or analyses are required to convincingly demonstrate this phenomenon.
>
>
> 4. Degradation in T5-XL
>
> For the difference between gains in T5-Large and T5-XL, I didn't compare only with HyperTuning.  For T5-Large, the proposed HART method surpasses all prior methods, including full-finetuning and PEFT, with the exception of FT-Few-Shot, which slightly exceeds HART. However, in the T5-XL case, FT-Zero-Shot, FT-Few-Shot, and HyperTuning-PT outperform HART. Additionally, Prefix-Tuning exhibits nearly identical performance to HART in this context.

---

### Official Review · Reviewer_hN68 · 2023-11-01

**Soundness:** 2 fair
**Presentation:** 2 fair
**Contribution:** 2 fair
**Rating:** 5
**Confidence:** 4

**Summary:**

The paper proposes HART, which improves HyperTuning with two components: autoregressive parameter generation and local consistency regularization. After training, the approach can generate the parameters of PEFT methods for unseen tasks without further tuning. The results show that the approach performs stronger than HyperTuning in some cases.

**Strengths:**

1. The approach considers the dependency of the generated parameters across layers, and this is a good observation in my opinion.

2. The performance of HART outperforms HyperTuning even though the latter approach requires further pre-training.

**Weaknesses:**

1. It's unclear to me (and somewhat counterintuitive) why using Fusion-in-Decoder performs better than the continual pre-training (the first and third row in Table 6), and have the authors explored the performance of HART + pre-training?

2. I am not fully convinced by the experiments that the methods are evaluated on the "unseen" tasks. In Tables 2 to 5, what is the separation between the training tasks and evaluation tasks? Are the tasks that appeared in training, not in the evaluation tasks? For example, question-answering is only used in training but not in evaluation. I found the paper didn't describe the dataset split in detail, but I think it is important in understanding the evaluation approach.

**Questions:**

1. In Table 3, do you have the accuracy of using FlanT5 to initialize the hypernetwork?

---

> ### Author Response · Authors · 2023-11-19
>
> We appreciate the reviewer for providing the helpful reviews. We provide our responses below.
>
> **It's unclear to me (and somewhat counterintuitive) why using Fusion-in-Decoder performs better than the continual pre-training (the first and third row in Table 6).**
>
> Fusion-in-Decoder (FiD) has demonstrated superior performance compared to the hypernetwork pre-training also in previous studies such as the HINT paper (Table 4, [[1]](https://arxiv.org/pdf/2212.10315.pdf)). The reason for its better performance might be attributed to the following:
>
> In FiD, we employ a direct concatenation of the output representation of the hypernetwork's encoder (referred to as "context") with the output representation of the main model's encoder (referred to as "query"). This fusion enables the main model's decoder to generate responses directly conditioned on the context. In contrast, in pre-training, the hypernetwork needs to learn the dependency between the context and query via learning a set of context-conditioned weights.
>
> Essentially, FiD simplifies the task for the model since ***it fuses context information into the main model more explicitly***, reducing the need for the model to learn context-query dependencies from scratch, resulting in improved performance.
>
> In our paper, we primarily adopt FiD because it offers comparable performance while incurring significantly lower computational overhead compared to pre-training. However, it's worth noting that both methods are complementary, as demonstrated in the HINT paper (Table 4, [[1]](https://arxiv.org/pdf/2212.10315.pdf)). If budget constraints were not an issue, we would employ both approaches to harness their combined benefits.
>
> [1] Ivison, Hamish, et al. "Hint: Hypernetwork instruction tuning for efficient zero-and few-shot generalisation."
>
> **Have the authors explored the performance of HART + pre-training?**
>
> We did not prioritize pre-training due to limited computing resources, as it is not the primary focus of this work. We plan to explore pre-training in future works on more advanced models than T5.
>
> We anticipate that pre-training will enhance the performance of HART. Given that HART relies on input dependencies between the hypernetwork and the main model, learning these dependencies in advance is expected to be beneficial.
>
> **I am not fully convinced by the experiments that the methods are evaluated on the "unseen" tasks. In Tables 2 to 5, what is the separation between the training tasks and evaluation tasks? Are the tasks that appeared in training, not in the evaluation tasks?**
>
> We strictly adhere to the training and evaluation subsets used in HyperTuning for both P3 and S-NI. Our evaluation tasks are entirely unseen during the training process.
>
> For P3, we use the T0 model's training subset, specifically selecting tasks with an average input length of fewer than 320 tokens following HyperTuning. We also use the T0 model's evaluation subset after excluding StoryCloze, as StoryCloze is not publicly distributed. The task taxonomy for both training and evaluation subsets can be found in Figure 2, Section 3 of the T0 paper [[2]](https://arxiv.org/pdf/2110.08207.pdf), which explicitly states, "To test zero-shot generalization, we hold out all constituent datasets... We also verify that data for those tasks is not leaked through the pretraining corpus." You can find the complete list of training tasks in Figure 6 of the HyperTuning paper [[3]](https://arxiv.org/pdf/2211.12485.pdf).
>
> For S-NI, we utilize the training and evaluation split from the Super-NaturalInstructions V2 dataset, excluding non-English tasks. The task taxonomy details are located in Section 5.1 and Appendix G of the S-NI paper [[4]](https://arxiv.org/pdf/2204.07705.pdf), where it states, "We split... into two subsets: one for evaluation and the other for supervision... To avoid data leakage, we exclude tasks from the training set if they are sourced from the same dataset as any test task."
>
> [2] Sanh, Victor, et al. "Multitask prompted training enables zero-shot task generalization."
>
> [3] Phang, Jason, et al. "Hypertuning: Toward adapting large language models without back-propagation."
>
> [4] Wang, Yizhong, et al. "Super-naturalinstructions: Generalization via declarative instructions on 1600+ nlp tasks."
>
> **In Table 3, do you have the accuracy of using FlanT5 to initialize the hypernetwork?**
>
> We do not have the accuracy of using FlanT5-XL to initialize the hypernetwork due to limited computing resources at this time. We hope to include this result in future for comprehensiveness. However, the gain on FlanT5-Large, which is ~6\%, should be sufficient to demonstrate the superiority of this initialization.
>
> ---
> We hope our responses address your concerns. We are happy to discuss your remaining concerns if any.

---

> > ### Comment · Reviewer_hN68 · 2023-11-21
> >
> > Thanks to the author for the response.
> >
> > I appreciate the author explaining the way to separate the tasks, but I would suggest including the task taxonomy in the next version so that readers can understand better about the setup.
> >
> > At the current content of the paper, the technical contribution is limited because most components are either used in HINT or HyperTuning (Table 1). The autoregressive generation part makes sense but only this contribution is not enough to make the paper big and novel. On the experimental side, I am still unsure if the method with pre-training can outperform HINT, and the current gap is not small either. Also, adding pre-training on top of HART would make the approach pretty similar to HINT, making the contribution unclear.
> >
> > Because of the above reasons, I will keep my original score.

---

### Meta-Review · Area_Chair_MU2S · 2023-12-17

**Metareview:**

This paper improves upon prior work of Hypertuning to efficiently predict parameter updates for tasks, without performing back propagation on task of interest. The paper does this via autoregressive parameter generation and local consistency regularization. The strengths are that the paper offers some empirical gains in settings studied. The reviews were all borderline, and the main weaknesses are the following:
(a) limited novelty - the paper currently mostly just combines two existing methods (as pointed out by multiple reviewers). This is fine, if it led to striking empirical gains. On the contrary, i think it adds more complex pieces that makes the method less likely to be adopted in practice
(b) significance - as reviewers have pointed out, the scalability of this approach is limited. Training a hypernetwork to predict parameters seems expensive (even if it is cheap when used)
(c) when does the method work - another concern was better taxonomy of how tasks are separated in train and test. How much do these networks actually generalize? It's not too useful if the generalization is limited to very similar tasks, especially if the hypernetwork requires large amounts of training.

There are also some unaddressed comments from a reviewer regarding clarity of exposition.

**Justification For Why Not Higher Score:**

Concerns above regarding novelty, significance and presentation.

**Justification For Why Not Lower Score:**

N/A

---

### Decision · Program_Chairs · 2024-01-16

Reject